# Refolding of Lysozyme in Glycerol as Studied by Fast Scanning Calorimetry

**DOI:** 10.3390/ijms23052773

**Published:** 2022-03-02

**Authors:** Alisa Fatkhutdinova, Timur Mukhametzyanov, Christoph Schick

**Affiliations:** 1A.M.Butlerov Chemical Institute, Kazan Federal University, Kremlevskaya 18, 420008 Kazan, Russia; alisamirovna@yandex.ru; 2Institute of Physics and Competence Centre CALOR, University of Rostock, Albert-Einstein-Str. 23-24, 18051 Rostock, Germany

**Keywords:** folding, lysozyme, glycerol, kinetics, intermediate, fast scanning calorimetry

## Abstract

The folding of lysozyme in glycerol was monitored by the fast scanning calorimetry technique. Application of a temperature–time profile with an isothermal segment for refolding allowed assessment of the state of the non-equilibrium protein ensemble and gave information on the kinetics of folding. We found that the non-equilibrium protein ensemble mainly contains a mixture of unfolded and folded protein forms and partially folded intermediates, and enthalpic barriers control the kinetics of the process. Lysozyme folding in glycerol follows the same or similar triangular mechanism described in the literature for folding in water. The unfolding enthalpy of the intermediate must be no lower than 70% of the folded form, while the activation barrier for the unfolding of the intermediate (ca. 140 kJ/mol) is about 100 kJ/mol lower than that of the folded form (ca. 240–260 kJ/mol).

## 1. Introduction

Seventy years since the first studies [1], protein folding is still one of the central topics of biophysics [2]. The term “the protein-folding problem” covers a broad range of questions, which can be broadly grouped into three key areas [3]: (a) study of the physical forces which encode the protein 3D structure, (b) revealing the reasons for the incredibly high folding rate, and (c) predicting the structure of the protein from the amino acid sequence. In the latter area, tremendous progress has been made in recent years. Modern computational approaches allow predicting the 3D structure of the proteins with a great deal of reliability [4]. However, understanding of how proteins fold and the physical forces behind the 3D structure of proteins is still being challenged [5]. This is partly due to the technical difficulties in monitoring and modeling the state of the protein while it folds. In typical cases, folding is a relatively fast process with a timescale of milliseconds to seconds, making experimental studies challenging, although the arsenal of experimental approaches to study protein folding is ever-expanding [6]. On the other hand, the large size of protein molecules restricts the application of computational techniques to model the folding process [7].

Early advances in the experimental observation of folding are related to applying rapid mixing techniques coupled to spectroscopic methods such as circular dichroism spectroscopy, fluorimetry [8], FTIR [9] or NMR [10]. In these approaches, folding is initiated by diluting a protein solution in the denaturing medium with water or a suitable buffer. As such, the environment of the protein molecules quickly changes from denaturing to native-like, and the native state becomes thermodynamically favorable. Thus, the non-equilibrium protein ensemble is produced and the development of the native structure is then observed via a suitable spectroscopic method. Careful analysis of the time evolution of protein spectra allows probing kinetics of the folding and even detects significant intermediates if present. A modification of this method enables performing temperature-jump experiments (both on heating and cooling) [9,11,12]. Recent advances include performing analysis of microdroplets using mass-spectroscopy [11].

While these approaches are undoubtedly fruitful, experimental investigation of folding remains challenging, and novel techniques may help to further the understanding of folding. Improtantly, the experimental study of folding to this day was mostly limited to the techniques which monitor the structural features of the protein molecules. As such, it is impossible to directly monitor the thermodynamic state of the protein ensemble during folding.

Differential scanning calorimetry is a valuable tool for analyzing protein thermodynamics [12,13]. However, the scanning rate of common DSC instruments is limited to about 1–100 K/min, and thus the timescale of the experiments (typically tens of minutes or hours per scan) is too long to monitor the state of the proteins during the folding events.

Fast scanning calorimetry [14] (FSC) is a novel experimental technique that makes use of MEMS-based chip-calorimeters [15] to achieve heating and cooling rates up to 10^6^ K/s [16]. Few applications of FSC analyses of proteins are described in the literature. Splinter et al. [17,18] used a liquid chip (a calorimeter chip with cover, allowing analyses of liquid samples at scanning rates up to 700 K/s) to study the denaturation of lysozyme, albumin, and blood serum.

Hen egg-white lysozyme is one of the most studied proteins. Several studies of the folding kinetics of lysozyme are described in the literature [10,19,20], including an investigation of the refolding in water-glycerol mixtures [21]. It was also shown that lysozyme correctly refolds in 99% glycerol [19,22], while later some differences in the conformations of lysozyme in water and glycerol were described [20]. Mukhametzyanov et al. [23] studied the denaturation of lysozyme in a glycerol solution at a heating rate of up to 8000 K/s on an open calorimeter chip. The glycerol solution was used to minimize solvent evaporation during the scan. It was noted that while the denaturation was found to be irreversible in the short time frame, a restoration of protein native structure over a longer time period occurred. However, in that work, the refolding process was not investigated.

In the present work, we used FSC to perform temperature-jump refolding experiments, and for the first time, perform the calorimetric analysis of the non-equilibrium protein ensemble on the example of hen egg-white lysozyme in glycerol.

## 2. Results

Fast scanning calorimetry operates following the same principles as differential scanning calorimetry. The minimization of the sample and the calorimeter itself by employing a chip-based measurement cell allows to reach high scanning rates both on heating and cooling. Due to the extremely small sample size, the surface to volume ratio is large and volatile solvents (such as water) can not be used. Thus, we have studied the refolding of hen egg-white lysozyme dissolved in the glycerol.

We performed temperature-jump experiments making use of the fast temperature changes possible with FSC (Figure 1).

The protein solution was first subjected to a series of rapid heating–cooling cycles (at 500 K/s) with an isotherm of 1 s from the instrument starting temperature (typically 25 °C) to 70 °C to ensure a stable baseline signal (segment 1). The initial protein solution may have residual water, which evaporates from the sample due to its small size (the typical volume of the sample droplet is ca. 10^−4^ µL) during these heating–cooling cycles.

After the stable baseline signal is achieved, the sample is heated to measure the unfolding effect of the protein (segment 2), then rapidly cooled to a temperature allowing refolding. After the sample is annealed at this temperature for a set time (segment 3), another heating scan (segment 5) is performed to analyze the calorimetric signal of the ensemble of protein molecules developed at the annealing stage. The additional cooling scan before heating (segment 4) is required when the annealing temperature is higher than the instrument starting temperature. The heating rates during the first and second scans were adjusted for each experiment between 10 and 2000 K/s depending on the sample size. The cooling rates after heating scans 2 and 5 were performed at 1000 K/s. The high cooling rates (>200 K/s) are not always possible with larger samples due to the thermal lag. However, as isothermal segments always follow the cooling scans, the sample has enough time to equilibrate thermally.

In fast scanning calorimetry, direct measurement of the sample mass is impossible. So, the first heating (segment 2) is not only used to generate the unfolded protein ensemble but also to register the temperature and the absolute enthalpy of the unfolding of the lysozyme in the sample.

After short annealing, the second scan does not contain any noticeable effects, while the baseline may be shifted due to the partial evaporation of glycerol at high temperatures (discussed later). However, after prolonged annealing, an endotherm at the same position as the initial effect of the unfolding begins to develop. Examples of such heating scans are shown in Figure 1.

After baseline correction, the comparison of the first and the second scans after annealing for 600 s reveals that the endothermal effect is almost completely restored in some experiments with a minor temperature shift (Figure 2). The restoration of the endotherm also means negligible contribution of aggregation, confirming the previous results where no aggregation was found in glycerol after even prolonged heating [23] using the dynamic light scattering technique.

At the intermediate annealing times, when the unfolding endotherm is not completely restored, an additional low-temperature endotherm is sometimes observed. An example is shown in Figure 3a; the comparison of the heating scans after baseline subtraction is shown in Figure 3b Note that the main peak retains a similar width and shape while having a lower height.

The separate integration of the peaks allows following the development of the peak areas. The peak areas from the second scan (segment 5) were normalized by the peak area from the first scan (segment 3) to compare different samples. It was possible to perform several measurements with the same sample by restoring the native protein content by 600 s annealing at 50 °C. To ensure a good signal-to-noise ratio, larger samples were used with the heating rate on the first and second scans of 100 K/s. The example of the progress curve of the peak area after annealing at 40 °C is shown in Figure 4.

The dead time for an experiment with a heating rate of 100 K/s is about 1 s, considering the time required to cool the sample from the annealing temperature to the instrument starting temperature, short isotherm before the heating for thermal equilibration, and the time needed to heat the sample from the starting temperature to the temperature of the first peak.

The progress curves of the main peak area were fit with the first-order kinetic equation to determine the apparent folding constant in pure glycerol. The Arrhenius plot of the apparent kinetic constant is shown in Figure 5.

To access the properties of the non-equilibrium protein ensemble, we have performed a series of scans with different heating rates in the second heating scan (segment 5 in Figure 1). The temperatures of the maximum of the endotherms depend on the heating rate (β). Evaluation of this dependence using the Kissinger plot (see [22] for details) yields the apparent activation energy of the processes responsible for the corresponding endotherm. Figure 6 shows the Kissinger plot of the peak temperatures of the main and additional calorimetric peaks.

## 3. Discussion

In the previous work [23], lysozyme unfolding in glycerol was studied using fast scanning calorimetry. It was found that the protein refolds when the sensor is left for a prolonged time at low temperature. In the present work, we study the kinetics of the refolding using the temperature–time profile that consists of two heating steps with an isotherm in between. The application of this profile allows to gain information on the thermodynamic state of protein ensemble during folding.

The endotherms observed during the heating scans of the proteins correspond to the uptake of heat required to reach the unfolded state with a higher enthalpy [24]. At higher scanning rates, the position of the endotherm is also determined by the kinetics of the unfolding [25]. The non-equilibrium protein ensemble contains a mixture of unfolded and folded molecules and possibly intermediate structures. If the non-equilibrium ensemble contains a large amount of partially folded molecules, the resulting curve is the sum of the contributions of such structures. The observed calorimetric curve contains two effects at intermediate annealing times, and the additional low-temperature effect can be attributed to the presence of the folding intermediate.

The shape of the calorimetric peak of unfolding of the partially folded forms can be estimated numerically on the basis of the transition state theory. Considering irreversible unfolding, the calorimetric curve of the unfolding of the partially folded intermediate can be described by the following equation [26,27,28]
(1)dHdT=ΔHIiUβd[Ii]dτ=−ΔHIiUβki[Ii],
where ΔHIiU is the unfolding enthalphy of the *i*-th intermediate, $k_{i}$ is the unfolding rate constant of *i*-th intermediate, $\left[I_{i}\right]$ is the concentration of the *i*-th intermediate, β is the heating rate during the scan, τ is time, and *T* is temperature. In a first approximation, the value of ΔHIiU can be considered temperature independent.

The temperature-dependent unfolding constant can be estimated on the basis of the transition state theory as
(2)$$k_{i}=\frac{k_{B}T}{h}\exp\left(\frac{\Delta S_{i}^{‡}}{R}-\frac{\Delta H_{i}^{‡}}{RT}\right),$$
where $\Delta H_{i}^{‡}$ is the enthalpy of activation of the *i*-th intermediate, $\Delta S_{i}^{‡}$ is the entropy of activation of the *i*-th intermediate, and $k_{B}$ and *h* are the Boltzmann’s and Planck’s constants, respectively. We may propose that there exists a set of folding intermediates with different values of ΔHIiU and the same transition state between the intermediate and the unfolded protein as between native and unfolded protein forms. The corresponding activation enthalpy will be lower by the enthalpy difference between the intermediate and the native protein:(3)ΔHi‡=ΔHN‡−ΔHNU−ΔHIiU=ΔHN‡−ΔΔHIiN,
where ΔHNU is the unfolding enthalpy of the native protein,$\;\Delta H_{N}^{‡}$ is the activation enthalpy of the native protein, and ΔΔHIiN is the difference between unfolding enthalpies of the *i*-th intermediate and native protein. We may also propose that the activation entropy of the intermediate is proportional to the activation enthalpy, and thus
(4)$$\Delta S_{i}^{‡}=\Delta S_{N}^{‡}\frac{\Delta H_{i}^{‡}}{\Delta H_{N}^{‡}},$$
where $\Delta S_{N}^{‡}$ is the activation entropy of the native protein.

The activation enthalpy and entropy of the native protein were estimated numerically and are equal to 228 kJ/mol and 370 J/mol·K, respectively.

The corresponding denaturation curves of the hypothetical partially folded intermediates are shown in Figure 7. As can be seen from the figures, the intermediates have lower peak temperatures. If the protein ensemble contains contribution from a number of various intermediate protein structures, the combined calorimetric curve is a sum of contributions of all present intermediates. Because of that, the shape of the peak can be expected to broaden and the temperature of the maximum may shift to lower temperatures.

However, in our experiments we only observed a peak with the same temperature and width as recorded during the first scan, and a single low temperature peak. When the main peak area is not restored completely, the position and the shape of the peak still remains identical to the first scan. The fact that the main effect retains the same width and shape means that the energy state corresponding to the folded structure is distinct. In other words, the next populated states have a rather large difference in the enthalpy and the unfolding barrier. As such, the kinetics of the process is determined by the enthalpic barriers.

The refolding of lysozyme in water proceeds over multiple pathways [10,29], and a triangular mechanism (off-pathway intermediate) was proposed as a minimum kinetic model (Figure 2) [29], where U is the unfolded protein, I is an intermediate, and N is the native (folded) protein. The formation of the intermediate appears to be guided by the disulfide bond configuration, as was shown by the molecular modelling study [30].

Using the annealing approach, we can verify that the refolding in glycerol follows the same scheme by refolding at elevated temperatures. The additional endotherm has a peak temperature of about 85 °C at a heating rate of 100 K/s. The temperature of 70 °C should be high enough to unfold the intermediate under isothermal conditions, so we can expect that the additional low-temperature endotherm is absent. Would the formation of the partially folded intermediate been the necessary step during the folding process, i.e., if the folding follows a sequential mechanism U⇄I⇄N, one expects that the high-temperature endotherm would not develop at all.

Contrary to that after annealing at at 70 °C (Figure 8), we observed the development of the high-temperature endotherm but not the low-temperature one. This experiment had to be performed at 1000 K/s heating rate with a smaller sample to avoid forming the intermediate during heating from the starting temperature of the scan (30 °C).

The development of the high-temperature peak in the absence of the low-temperature peak means that there is a direct pathway for folding not involving a formation of the intermediate.

As visible from Figure 5, the temperature dependence of the apparent refolding constant follows the Arrhenius behavior between 10 and 50 °C with the apparent activation energy of 92 ± 6 kJ/mol. However, between 50 and 70 °C, the temperature behavior is anti-Arrhenius.

This can be explained by the consideration of the kinetic model proposed in [29]. At temperatures below at least 70 °C, the folded lysozyme form appears to be the equilibrium one. Thus, the reactions leading to this form can be considered irreversible (the kinetic constants leading to the folded form are far greater than the constants of the reverse processes). The formation of the intermediate, however, may be controlled by a reversible reaction with a corresponding equilibrium constant. We consider the process to be quasi-equilibrium; that is, the equilibrium between unfolded and intermediate form is established faster than the rates of the processes leading to the folded form. The simplified scheme for the process is then as follows (Figure 3):

As such, the formation of the folded form is approximated by the following differential equation:(5)$$\frac{dN}{d\tau }=k_{1}U+k_{2}I=k_{1}U+k_{2}K_{I}U=\left(k_{1}+k_{2}K_{I}\right)U.$$

If the conversion of the intermediate to the folded form has a large contribution to the overall rate of formation of the folded form (that is $k_{1}$<$k_{2}$) than at the temperatures where the intermediate is already thermally unstable ($K_{I}$→0), the temperature dependence of the folding rate may appear anti-Arrhenius.

That said, the direct folding pathway may also be a multistage one and itself may contribute to the anti-Arrhenius behavior.

Finally, we may try to characterize the properties of the intermediate form responsible for the low-temperature endotherm. Unfortunately, the molar enthalpy of the unfolding of the intermediate cannot be calculated from the available data, as to perform that, we need to know the relative amounts of the intermediate and folded forms, which is impossible since the unfolded form does not register in the heating scan. Moreover, the Van’t Hoff relation cannot be used to estimate the enthalpy of the intermediate, as its unfolding during the heating scan proceeds under non-equilibrium conditions due to the high heating rate. However, a lower bound can be put on the enthalpy of the unfolding of the intermediate.

If we consider the total amount of protein molecules to be 1, then at any given moment N+I+U=1.

Thus, the following inequality must be satisfied:(6)ΔH1ΔhNU≥ΔH2, mainΔhNU+ΔH2, addΔhIU,
where $\Delta H_{1}$ is the absolute enthalpy of the endotherm recorded on the first heating scan, $\Delta H_{2,\;main}$ is the absolute enthalpy of the main peak recorded during the second heating scan, $\Delta H_{2,\;add}$ is the absolute enthalpy of the second (low-temperature) peak, ΔhNU is the molar enthalpy of the unfolding of the folded protein form, and ΔhIU is the molar enthalpy of the unfolding of the folded intermediate. From inequality (6), it follows that ΔhIU must have a lower bound; otherwise, the inequality would not hold.

The evaluation of the experimental data gives a safe estimate of ΔhIU to be no less than 70% of ΔhNU.

We also may estimate the activation parameters of the unfolding of the intermediate from the Kissinger plot. For the main peak, we get the apparent activation energy of the corresponding process to be 240 ± 29 kJ/mol, which agrees with the previous estimate [23], and for the low-temperature peak, we get the apparent activation energy of 142 ± 14 kJ/mol. Furthermore, the numerical fit of the low-temperature endotherm produces a value of activation energy of 139 kJ/mol and a good quality fit (Figure 9). In terms of the transition state theory, the activation enthalpy of the intermediate is 136 kJ/mol and the activation entropy is 174 J/mol·K.

Thus, the energy barrier for the unfolding of the intermediate form is about 100 kJ/mol lower than that of the final folded form.

Finally, we may discuss the effect of the neat glycerol, as a solvent, on the folding process. It must be noted, that in contrast to previous studies of the effect of glycerol on lysozyme [20,21,31,32,33,34] and other proteins, e.g., [28], the protein environment after preheating is dry glycerol. This is certainly true after the first heating when the sample reaches 140 °C. Surprisingly, the obtained results show a remarkable similarity to the previous results obtained in water [10,19,20]. Thus, it seems that the process of folding of lysozyme in glycerol follows conceptually the same pathway(s) as in water. As the solvent environment of lysozyme is pure glycerol, effects of preferential solvation which affect the stability of proteins in water-glycerol solutions can be excluded. At the same time, the apparent folding constants are much lower in glycerol, e.g., the apparent folding constant at 20 °C in glycerol is about 500,000 times lower than that in water. As found in the previous work [23], such parameters as enthalpy of unfolding and apparent activation energy of unfolding of lysozyme are similar in water and glycerol. A much lower refolding rate indicates lower molecular mobility of the protein chain and might be a result of the much higher viscosity of the glycerol.

In closing, we would like to discuss the outlook for the application of FSC in protein unfolding/folding research as compared with the conventional DSC. First, we have to again mention that the application of the open calorimetric sensors is limited to virtually non-volatile solutions. To our knowledge, glycerol is the only solvent which is able to support native-like protein structure (at least of certain proteins) and has low volatility. Second, the determination of the molar calorimetric enthalpy of the unfolding, using FSC, is complicated, though possible (example and discussion is available in the previous work [23]). Closed calorimetric sensors are available and some applications were described in the literature [17,18].

The faster scanning rates, available in FSC, not only allow to perform temperature-jump experiments to study the protein ensemble during folding, but have other important advantages, as well. The high heating rates limit the duration of the exposure of the protein solution to the high temperatures, thus decreasing possible contribution of the irreversible chemical degradation (e.g., deamidation [35]) or aggregation. Of course, if the protein solution is then rapidly cooled, the aggregation of the unfolded protein may occur (we have not detected it in the present work). If aggregation prevents refolding, the endotherm of the native structure does not restore in the second scan. If the aggregates dissociate with a heat uptake upon heating, the corresponding thermal effect can, in principle, be detected, provided the sensitivity is enough.

Another advantage of FSC technique might be the small sample size, however, the protein solution must be fairly concentrated because the sensitivity of the calorimetric sensor is limited.

It must also be noted that due to the high heating rates available in FSC technique, the endotherm shape is determined by the kinetics of unfolding. Thus, the technique is sensitive to the changes in the protein structure (such as presence of the disulfide bonds) that affect the unfolding kinetics.

Finally, while there are differences between the experimental setup and the information available from the FSC as compared with the regular DSC, the former is a subset of the latter, and thus FSC may be applied (considering its limitations) to the same objects, including different types of protein structures.

## 4. Materials and Methods

We have used hen egg-white lysozyme (lyophilized, #62970, St. Louis, MO, USA) and glycerol from Sigma (>99%) without further purification. A solution containing 50 mg/mL of lysozyme was prepared; the rather high protein concentration is required to obtain a good calorimetric signal-to-noise ratio. The lysozyme solution was prepared using the same protocol as previously used [23] by dissolving the lyophilized protein with the corresponding amount of neat glycerol. To facilitate dissolution, the glycerol was heated to 50 °C and the mixture was sonicated before the clear solution was obtained. The lyophilized protein contains ca. 10% water by weight, thus the resulting solution may contain approximately 0.5% water by weight. This residual water is eliminated by performing a series of heating and cooling scans on the sample before the main heating scans.

The experiments were performed using a Flash DSC 2+ (Mettler-Toledo, Greifensee, Switzerland) with a UFS1 sensor, calibrated with biphenyl and benzoic acid melting effects as temperature standards.

In the typical experiment, a droplet of the lysozyme solution was placed on the center of the UFS1 sensor. The size of the droplet was chosen to both minimize thermal lag and maintain a good signal-to-noise ratio. Thus, a bigger droplet was used for relatively slow scanning rates, ca. 100 K/s, and smaller droplets were used for faster scanning rates, ca. 1000 K/s.

Depending on the size of the sample, it is possible to perform single or multiple measurements with one and the same droplet. Glycerol can evaporate from the smaller samples and a noticeable mass loss was detected after performing two heating scans to 140 °C according to the temperature–time profile from Figure 1, so a new sample was used for each measurement. However, because the mass and the geometry of each new sample is different, the reproducibility of the kinetic experiments was poor. Larger samples could be reused several times with 600 s annealing at 50 °C between measurements to restore the concentration of the folded protein. While some glycerol evaporation occurs, due to larger droplet size the relative mass loss is low, and the area and temperature of the denaturation peak after long annealing are close to the first recorded scan from the same droplet. This improves the reproducibility of the kinetic experiments. To improve the resolution and the quality of the baseline, a droplet of pure glycerol of comparable size was placed on the reference side of the sensor.

## 5. Conclusions

In the present work, we have studied the folding of lysozyme in glycerol using fast scanning calorimetry. FSC provides a unique view of the properties of the non-equilibrium protein ensemble as it directly probes the energetics of the sample.

We have found that the kinetic mechanism of folding in neat glycerol is similar to that described for water solutions. The calorimetric curves of the non-equilibrium protein ensemble reveal that the only protein states which produce distinct calorimetric peaks are those of the native structure and a folding intermediate. Thus, apparently, the kinetics is controlled by the enthalpic barriers; once protein molecule crosses that barrier it rapidly assumes a compact cooperative structure. Folding occurs via different pathways, with a partially folded intermediate forming in one of the routes. The temperature dependence of the apparent kinetic constant of the refolding is non-Arrhenius in the region between 50 and 70 °C, which may be explained by the marginal thermal stability of the partially folded intermediate.

Unfortunately, it is impossible to measure the unfolding enthalpy of the partially folded intermediate, but a lower estimate of 70% of the value of the folded (ca. 550 kJ/mol for the folded form and >380 kJ/mol for the intermediate) form can be given.

The apparent activation energy of the partially folded intermediate (142 ± 14 kJ/mol) is about 100 kJ/mol lower than that of the folded form (240 ± 29 kJ/mol found here and 261 ± 13 kJ/mol reported earlier under similar conditions in [23]).

The kinetics of folding in glycerol is much slower than in water, which probably is the result of the high solvent viscosity.

The results obtained in the present work contribute to the better understanding of the physical forces which drive folding and to understanding folding kinetics.

## Data Availability

The data presented in this study are available on reasonable request from the corresponding authors.

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
