# Peer review of "Refolding of Lysozyme in Glycerol as Studied by Fast Scanning Calorimetry"

_ijms, 2022, doi:10.3390/ijms23052773_

Round 1

Reviewer 1 Report

Title: Refolding of lysozyme in glycerol as studied by Fast Scanning Calorimetry

Journal: International Journal of Molecular Sciences

In this research, the authors studied the folding of lysozyme in glycerol using fast scanning calorimetry. The authors examined the heat flow during lysozyme thermal folding and extracted the folding pathway and activation energies. The folding pathway of lysozyme has been well-established during the past decades. The authors hope to provide a view from the energetic information.

Major concerns:

  1. The writing of the manuscript should be improved. Some paragraph in “2. Experiment” seems to be the establishment of protocol or the verification of the experimental procedures.
  2. I am not sure about the novelty of this work. Any technical improvements? Any new information for the well-studied model protein lysozyme?
  3. It seems that the possible model of lysozyme thermal folding was deduced from the authors’ explanation of the experimental data. In this case, it is difficult to say the evidences were solid, but was more like to be deduced from previous observations. Therefore, I can hardly see any new information.
  4. The short explanation of the effect of glycerol seems too preliminary and simple. It has long been known that glycerol affects protein folding, at least partly due to its osmotic effect. I can not agree with that the difference was caused by viscosity. Actually, the differential physical-chemical properties between glycerol and water makes the solvent molecules interact with the protein molecules differentially, which further leads to different free Gibbs energies of the native and denatured states. (please refer to the numerous papers describing the osmotic effects of glycerol, for example, Meng et al., Biophys J. 2004 Oct;87(4):2247-54. doi: 10.1529/biophysj.104.044784; Feng et al., 2008 May 1;71(2):844-54. doi: 10.1002/prot.21744.)

Minor comments:

  1. Page 1, last paragraph: I do not think NMR is frequently used as kinetic folding studies for fast-mixing techniques. NMR can study molecular dynamics by measuring specific dynamic-related parameters. Generally, fluorescence, CD, UV-vis absorbance as well as vibrational spectra including FTIR are used for stop-flow or fast kinetic studies, esp. for the studies in the old days.
  2. Figure 1, it is better to rename Figure 1 as Scheme 1 since it is not a figure generated from experimental data but is more like an experimental protocol or scheme. I am not sure about the usage of the word “section” (“segment” was used somewhere, please keep these constant throughout the manuscript). Maybe “period”, “phase” or something like this be more appropriate? Furthermore, maybe “protocol” or “procedure” is better for “program”? The later two suggestions are just suggestions but not mandatory.
  3. The unit of “heat flow” for all figures?
  4. Page 7 and the following pages, please name the figures continuously.
  5. Page 7, “Figure 1”, please add the details of simulations in the legend.
  6. Page 8, line 206, “As visible from Error! Reference source not found.”?
  7. The so-called triangle model seems to be off-pathway intermediate model.

Reviewer 2 Report

Fatkhutdinova et al. reported an experimental research on the folding of lysozyme in glycerol using the Fast Scanning Calorimetry technique. They discovered that lysozyme folding in glycerol similar to the folding in water, and provided the insights on the properties of the non-equilibrium protein ensemble. It is an interesting paper, however, the interpretation of the data should be substantially improved. Here are my concerns.

  1. Please clarify what we can learn from this lysozyme study in a general view of protein folding and unfolding?
  2. Why did the authors use glycerol to do the protein folding study?
  3. The author mentioned three key areas for protein folding study. After reading the paper, I am not sure which one the current paper focuses. Please clarify it.
  4. I encourage the authors to compare the current experimental data with  molecular modeling analysis to provide insights for the related area.
  5. The protein folding studies on lysozyme have been conducted for years. Please compare the current study with those works and highlight the novelties.
  6. The figure numbers are duplicated in 1, 2, 3.

Reviewer 3 Report

In this manuscript authors have studied folding of lysozyme in glycerol using  Fast Scanning Calorimetry technique. Folding/unfolding of lysozyme is an important topic of study and has got much attention from the researchers working in protein interactions. Although the work related to the lysozyme folding/unfolding/refolding has already been reported many times using calorimeters and other potential techniques, authors claim that they have applied fast scanning calorimetry using rather new types of MEMS-based chip-calorimeter. The main drawback of this work is that it lacks the comparison with the similar works reported previously using classical methods and why the authors have chosen this topic and which advances have been left in these works. Furthermore, in an article published by some of the authors of this manuscript that is related to the unfolding/denaturation of lysozyme in glycerol using fast scanning calorimetry, authors discussed briefly about the refolding of lysozyme, thus, it is recommended that the authors discuss the present findings on the basis of the discussion giving in their previously published work (Fast scanning calorimetry of lysozyme unfolding at scanning rates from 5 K/min to 500,000 K/min, Biochimica et Biophysica Acta (BBA) - General Subjects Volume 1862, Issue 9, September 2018, Pages 2024-2030).  The other minor point is that kindly describe the methods clearly in the manuscript rather to give the reference of any other published paper. The comparisons of the conclusions obtained in the previous works on the same topic should also be elaborated. 

Reviewer 4 Report

In the manuscript ijms-1588710 Refolding of lysozyme in glycerol as studied by Fast Scanning Calorimetry, Fatkhutdinova et al study lysozyme folding using Fast Scanning Calorimetry (FSC). The authors use temperature-induced unfolding and refolding of the protein to successfully assess the energetics, and indicate the temperature dependant kinetic constants. The article is well written and the authors have described the technique with clarity. Listed below are a few suggestions, comments, and questions for the authors.

1) A head-to-head comparison of another approach to study lysozyme folding/refolding is required here in order to prove the superiority of FSC over other approaches used to address the same question on folding. Also, at least one measurement should be carried out in the water or a suitable buffer to compare the folding kinetics against glycerol. Studying protein folding under such conditions would also improve physiological importance to the work

2) Can the native state of the lysozyme be exclusively tracked using the method described in this work? Please explain and/or add relevant data

3) Using FSC, is it possible to study events like protein aggregation and disaggregation? Please include a concise account of this idea/possibility

4) How would lysozyme under slow folding conditions, with a relatively higher number of intermediate states, behave compared to a fast-folding track when monitored using FSC?  This could be done perhaps by either using different solvents that might lead to a dissimilar folding landscape or utilizing the same solvent but adjusting appropriate parameters to alter the folding kinetics

5) As the authors point out correctly, glycerol, due to its viscosity, might slow down folding kinetics. Taking this feature as an advantage, the authors should be able to measure free energy differences during folding. Please include this data

6) How will the endotherms vary in proteins having disulfide bonds compared to ones that do not? Please discuss briefly.

Round 2

Reviewer 1 Report

Although I still have some concerns about the novelty (the model is the same as those defined in water), the authors have addressed most of my previous concerns.

Please carefully check whether there are formating problems since some words/citations are irrecognizable in the version for the reviewer that I downloaded. Maybe the formating problems could be solved by the help of the journal editors.

Reviewer 3 Report

Authors have revised the manuscript properly and it is acceptable in the present form.

Reviewer 4 Report

In the revised version of the manuscript, Refolding of lysozyme in glycerol as studied by Fast Scanning Calorimetry, the authors have attempted to improve the discussion to address some questions raised by the reviewer. Given the limitation of not having a closed calorimetric sensor in this study, the authors were unable to answer a few other questions; and the authors indicate their plans to pursue relevant studies in the near future to circumvent the shortcomings of the current method they have used. The manuscript may be accepted for publication in its current form.
